# Collaborative Training of Balanced Random Forests for Open Set Domain Adaptation

## Abstract

In this paper, we introduce a collaborative training algorithm of balanced random forests for domain adaptation tasks which can avoid the overfitting problem. In real scenarios, most domain adaptation algorithms face the challenges from noisy, insufficient training data. Moreover in open set categorization, unknown or misaligned source and target categories adds difficulty. In such cases, conventional methods suffer from overfitting and fail to successfully transfer the knowledge of the source to the target domain. To address these issues, the following two techniques are proposed. First, we introduce the optimized decision tree construction method, in which the data at each node are split into equal sizes while maximizing the information gain. Compared to the conventional random forests, it generates larger and more balanced decision trees due to the even-split constraint, which contributes to enhanced discrimination power and reduced overfitting. Second, to tackle the domain misalignment problem, we propose the domain alignment loss which penalizes uneven splits of the source and target domain data. By collaboratively optimizing the information gain of the labeled source data as well as the entropy of unlabeled target data distributions, the proposed CoBRF algorithm achieves significantly better performance than the state-of-the-art methods. The proposed algorithm is extensively evaluated in various experimental setups in challenging domain adaptation tasks with noisy and small training data as well as open set domain adaptation problems, for two backbone networks of AlexNet and ResNet-50.

## 1 Introduction

In recent years, domain adaptation has been researched as it can help to solve major difficulties in the real world. Due to the huge overhead in labeling large-scale training data, it is desirable if an existing network can be adapted to different target domains. More importantly, it is common that the training dataset for adaptation is noisy and small, or the labels in the target domain do not match with the source or even unknown. These are inherent challenges in the domain adaptation problem as in real world it is common for the data to contain such class bias, noise and unlabeled data.

However, in practice, since the adapted networks are often overfitted to the provided source data or the data distribution of the target domain is frequently quite different from the source, they do not perform well to the target domain. To properly deal with these real-world conditions with insufficient information, it is critical to learn the shared data distribution that is effective both in the source and target domain. To this end, we propose the collaborative training algorithm of balanced random forest (CoBRF) to mitigate the challenging problems such as noisy labels, lack of training data, and misaligned or unknown categories (open set categorization).

In random forests, multiple decision trees are learned by optimizing the information gain for the randomly selected subset features at each node split. Since random forests ensemble the internal decision trees, they are more robust to noise and overfitting problem than single decision trees. To improve the robustness of the random forests, we take one step further by balancing the decision trees, i.e., maximizing the number of leaf nodes for the same tree depth. Our method builds more balanced decision trees by enforcing the sizes of the data in the left and right child nodes to be equal. While this split strategy is not locally optimal in terms of information gain, the resulting decision trees have far more leaf nodes, and it endows more expressive power which can be helpful in dealing with noise and unseen data or classes. It also helps to avoid overfitting as it prevents a node committing too early

for a specific pattern, or in other words, it postpones the decision as late as possible so that various discriminant information in the training data can be fully considered.

To enforce even splits while maintaining the discriminability, the CoBRF uses the hyperplanes estimated by the linear support vector machine (SVM). First, it randomly assigns the classes in the nodes to binary pseudo labels and equalizes the sizes of two pseudo classes by randomly removing data in the larger class. Then a linear classifier is found by SVM, and its hyperplane is translated until the data sizes on both sides are equal. In a sense, it finds the even split of the data projected onto the normal direction of the hyperplane and places the hyperplane there. The node split by the translated hyperplane is simple yet effective. The ablation study in Sec. 4.2 confirms that the CoBRF boosts the performance compared to the baseline random forests.

Since the above training process only considers maximizing the information gain of labeled data in the source domain, which is referred to '**class information gain**', it does not resolve the domain misalignment problem between the source and target domain. Because the target labels are not available during training, we try to keep the overall distribution of the target data as close to that of the source data as possible. Since the source data are evenly split, we guide the algorithm to minimize the information gain between the source and target domain, which encourages even split of the target data also. The CoBRF combines the ideas, minimizing the '**domain information gain**' between source and target data for the domain alignment while keeping the **class information gain** to be maximized. Note that the domain alignment term is the same as the negative information gain of the binary domain labels (source/target). Thus, the CoBRF can be seen as an example of adversarial learning, as it considers the domain information gain in an adversarial manner compared to the conventional objective function of the random forest.

We summarize the main contributions as three-fold.

- We introduce the collaborative training algorithm based balanced random forest (CoBRF) using the discriminative and even node split function. Linear SVM with binary pseudo labeling is used to find the discriminative hyperplane and the even split ensures the decision tree to be balanced.

- We also adopt the adversarial learning of domain information gain to align the source and target data distribution. To align two domains, the information gain between the source and target data is minimized, which learns the common data distribution of both the (unlabeled) target domain and the source domain data.

- We perform an extensive evaluation of the domain adaptation to show the performance of the proposed method according to various challenging evaluation protocols. Specifically, it is compared to the baseline and state-of-the-art methods using noisy and small training data, and with open-set domain adaptation protocols. In both cases we observe significant performance improvements.

## 2 RELATED WORK

### 2.1 DOMAIN ADAPTATION

Recently adversarial learning has been one of the dominant approaches in domain adaptation with deep neural networks. The gradient reversal layer Ganin & Lempitsky (2015) is introduced to train the networks so that the discrimination of source and target domains is penalized. It improves the classification performance compared to the networks learned only with the source data. Tzeng et al. (2017) suggest the domain adaptation framework based on the discriminative network learning, which assigns individual weights to the source and target domains. In training the networks, they also consider the adversarial weight update to align the domains. Several other domain adaptation papers in adversarial learning using conditional learning Long et al. (2018), domain-symmetric Zhang et al. (2019), and collaborative Zhang et al. (2018b) methods have been introduced. Also, in Tzeng et al. (2014); Long et al. (2015; 2016), maximum mean discrepancy (MMD)-based methods have been studied. Tzeng et al. (2014) propose the domain confusion loss to improve domain distribution alignment. Long et al. (2015) introduce the task-specific embedding and multiple kernel approach along with MMD to decrease the domain discrepancy. The residual transfer module presented in Long et al. (2016) associates the classification ability of the source and target domain. MMD is further extended to multiple domain alignment in the joint adaptation networks (JAN) Long et al. (2017) using adversarial learning. The generative adversarial networks Radford et al. (2015) are adopted in

many domain adaptation methods Liu & Tuzel (2016); Sankaranarayanan et al. (2018); Volpi et al. (2018). CoGAN proposed by Liu & Tuzel (2016) learns the joint distribution of multiple domains without corresponding image pairs. Sankaranarayanan et al. (2018) propose the combined adversarial and discriminative learning method using the generator and discriminator of GAN.

## 2.2 Evaluation Protocols in Domain Adaptation

Recently, many challenging protocols are introduced to evaluate the domain adaptation in realistic settings. Regarding domain generalization on deep neural networks Li et al. (2018); Balaji et al. (2018), they divide multiple domain data into training and test set, then use the leave-one-domain-out scheme for evaluation. The domain adaptation on the partially overlapping source and target domains is presented in Zhang et al. (2018a); Cao et al. (2018). Multiple sources and target domains are mixed into the source or target domains in Zhao et al. (2018); Mancini et al. (2018); Hoffman et al. (2018). The adaptable model is aimed to be learned using the distribution to the multiple domains of the mixed set. Recently, several works Saito et al. (2018); Panareda Busto & Gall (2017); Tan et al. (2019) address the open set domain adaptation. They assume that there exist unknown and partially overlapped known classes between domains. On the other hand, the domain adaptation methods under small training data Hong et al. (2017) and the noisy data Shu et al. (2019) are studied to address the real-world condition. Hong et al. (2017) use single training data per person, and Shu et al. (2019) artificially corrupt the class labels or features of the source domain for the robustness evaluation. These protocols are challenging as they pose difficult problems of overfitting, class misalignment, noisy, lack of training data, and little overlap.

## 2.3 Random Forest as an Ensemble Learning Method

The ensemble of multiple learners has widely been used to avoid the overfitting problem Singh et al. (2016); Han et al. (2017; 2016); Pi et al. (2016). Singh et al. (2016) introduce the regularization method for network learning, which works with a variety set of network architectures and performs better than the existing regularization methods (*i.e.*dropout). Branchout Han et al. (2017) is devised for layer-level regularization in visual tracking, where multiple branches of fully connected layers are randomly selected in training.

Random forest Breiman (2001) combines multiple random decision trees to build robust classifier or regressor. Random forests have been applied to many applications such as object tracking Zhang et al. (2017), feature point detection Lindner et al. (2014), and speech recognition Black & Muthukumar (2015), to name a few. However, it should be emphasized that the most important benefit is the mitigation of overfitting by ensembling multiple decision trees. As noticed in the literature Wyner et al. (2017); Gomes et al. (2017), the random forests tend not to propagate severe overfitting error even with a large number of trees.

There have been many recent works to improve the performance of random forests: Dheenadayalan et al. (2016) proposes pruning nodes for efficient learning, Ristin et al. (2015a) presents incremental modeling for large scale recognition, and Probst & Boulesteix (2017) investigates how to tune the number of trees. SVM Yao et al. (2011); Ristin et al. (2015b) or random projection Bosch et al. (2007); Bossard et al. (2014) is often used as the binary classifier for better node split. Training balanced decision trees has been also an important topic Bosch et al. (2007); Bossard et al. (2014); Yao et al. (2011); Lei et al. (2014); Ristin et al. (2015b). They split a node into child nodes by the binary classifier, which is trained by evenly-divided training data in the node. We argue that training balanced random forests helps to alleviate the overfitting problem since balancing random decision trees avoids the biased distribution in the specific domain but prefers the common representation to any domains. Hence, we introduce the learning algorithm that enforces the even-split constraint by shifting the hyperplane(Sec. 3.1) for balanced random forests. Although there have been studies of the balanced training of random forests, we provide elaborate training process of balanced random forests to learn common representations for the domain adaptation task. The effectiveness of the balanced random forests is shown by extensive domain adaptation experiments.

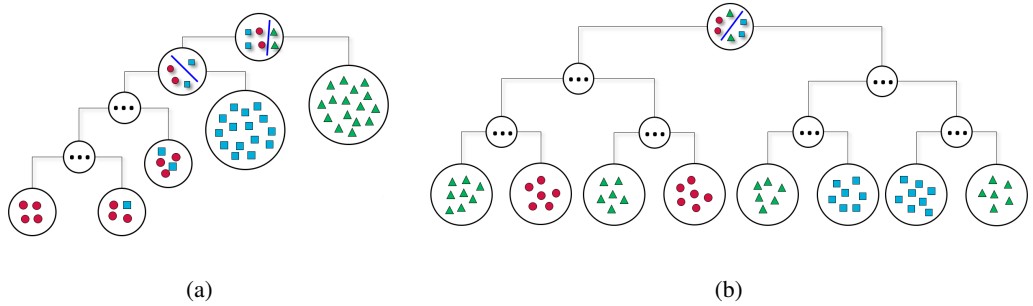

(a)          (b)

Figure 1: Split examples in decision tree according to the split functions: (a) the conventional method chooses the split that maximizes the information gain. (b) In contrast, the proposed method additionally enforces the size of child nodes to be equal, resulting in a random balanced tree. Note that CoBRF has far mode nodes which improves the generalization ability for domain adaptation.

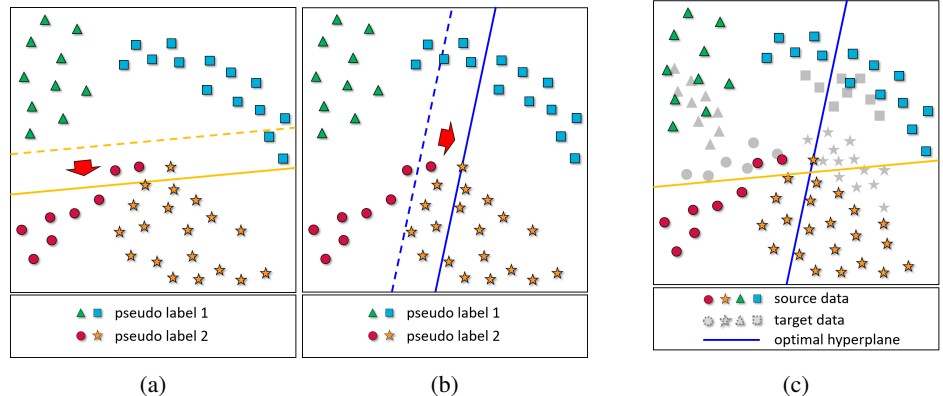

(a)          (b)          (c)

Figure 2: Hyperplanes by the proposed methods. (a,b) The hyperplanes estimated by binary pseudo labels followed by translation for even split. Dotted line is the hyperplane estimated using linear SVM. The data are evenly split by the hyperplane shift (solid lines). Among these hyperplanes, the one with maximum information gain is chosen: yellow hyperplane in (a). (c) In CoBRF, both the source information gain and target entropy is considered. The yellow is better in source information gain, the target data split is biased, while the blue splits the source and target evenly well.

## 3   PROPOSED METHOD

In this section, we first explain the limitation of the conventional random forests for the domain adaptation task, and then we introduce the even node split function in Sec.3.1 and the domain information gain for selecting the domain-aligned split function in Sec. 3.2.

### 3.1   EVEN CONSTRAINED RANDOM FOREST LEARNING

A random forest consists of multiple random decision trees, whose nodes learn a binary classifier for the randomly-selected subset of features to maximize the information gain (IG). We abuse the term node for the training data in the node interchangeably. The entropy of a node $n$ is defined as

$$E_{\mathcal{C}}(n) = - \sum_{c \in \mathcal{C}(n)} p_c(n) \cdot \log\left(p_c(n)\right), \tag{1}$$

where $\mathcal{C}(n)$ represents the set of classes of the data in $n$, and $p_c(n)$ is the probability of class $c$ in $n$ (i.e., the data count of class $c$ divided by $|n|$). Then the information gain for a node $n$ with the left

and right child nodes is defined as

$$IG_\mathcal{C}(n) = E_\mathcal{C}(n) - \sum_{l \in \{left, right\}} \frac{|n_l|}{|n|} E_\mathcal{C}(n_l). \tag{2}$$

Conventionally, the simple split functions that compares only a couple of feature values are used, but recently more elaborate split functions using the linear classifiers are used Yao et al. (2011); Ristin et al. (2015b). The hyperplane split function for a node $n$ is written as

$$\nu_n(\mathbf{x}) = \begin{cases} \text{go left,} & if \ \mathbf{w}_n \cdot \psi_n(\mathbf{x}) < k_n \\ \text{go right,} & otherwise, \end{cases} \tag{3}$$

where $\psi_n(\cdot)$ is the sub-feature selection function and $\mathbf{w}_n$ and $k_n$ are the hyperplane parameters either randomly set or learned by a linear support vector machine Cortes & Vapnik (1995). The hyperplane with the largest information gain is the most discriminative classifier at the given node, but for the entire decision tree and the random forest it may not be the best option, because it causes the learned trees to be skewed and not well balanced (Fig. 1a).

We propose to add a hard constraint of equal-size in splitting the node to get more balanced trees. The detailed learning process is as follows. For the SVM to build a binary classifier, the classes in the node are randomly assigned to binary pseudo labels, and the training data for each class are assigned to the corresponding pseudo label. As the data sizes of the pseudo labels will be different, we randomly erase the data in the larger pseudo class to match the sizes. Then the base hyperplane ($\mathbf{w}_n$ and $k_n$) is computed to classify the binary pseudo labels.

Still, the split of the training data by the hyperplane is not equal-sized; thus we update the bias $k_n$ of the hyperplane so that the data size on each side is equal or differs at most by one ($||n_{left}| - |n_{right}|| \leq 1$). Geometrically this process is moving the hyperplane along the normal vector $\mathbf{w}_n$, so that it is placed at the even split of the data projected onto the normal direction (Fig. 2a,2b). Among the estimated hyperplanes from randomly selected sub-features, the one that maximizes the information gain $IG_\mathcal{C}(n)$ is chosen as the node split function. To build a decision tree, like the conventional random forest, the node split is repeatedly applied until the maximum depth is reached or too few data are left in the node (Fig. 1b).

Inherently the proposed split method creates balanced trees, and for the same depth, the number of nodes is much larger than that of the conventional random forest. We argue that having more (leaf) nodes in the decision tree has advantages in domain adaptation tasks. The conventional split function is locally optimal, but because of that, it is more susceptible to overfitting by committing too early, and eventually, it decreases the discriminative power of the entire random forest. In the balanced trees, the data sizes in the leaf nodes are almost the same; thus, they represent local data distribution more faithfully. The even-size constraint can be thought of as a regularization in learning decision trees. The experimental results of the ablation study in Sec. 4.2 supports this argument.

## 3.2 Collaborive Learning of Random Forests

Balanced data distribution is a big advantage in domain adaptation. However, as it does not use the unlabeled target data for learning, it still does not correctly align the data distribution of the source and target domain. In other words, the distribution of the target data also needs to be considered in building a random forest. We propose a new collaborative measure for selecting the split function that considers both the conventional IG and the domain distribution of the source and target data together. The collaborative information gain (co-IG) is defined as

$$co\text{-}IG(n) = (1 - \lambda) \, IG_\mathcal{C}(n_s) - \lambda \, IG_\mathcal{D}(n), \tag{4}$$

where $\lambda$ is a user parameter, $n_s$ is the labeled training data (in the source domain), and $\mathcal{D}$ is the binary domain label $\{source, target\}$ representing the domain that the data belongs to. More specifically, $IG_\mathcal{D}(n)$ is the information gain on the domain distribution, when the data labels are either source or target, disregarding the classes in the source domain. The CoBRF chooses the hyperplane that maximizes co-IG when splitting the nodes.

Note that the IG on the domain distribution, $IG_\mathcal{D}(n)$, is subtracted in Eq. 4, to ensure that we prefer even distribution of the source and target data in the child nodes. $IG_\mathcal{D}(n)$ is minimized when both

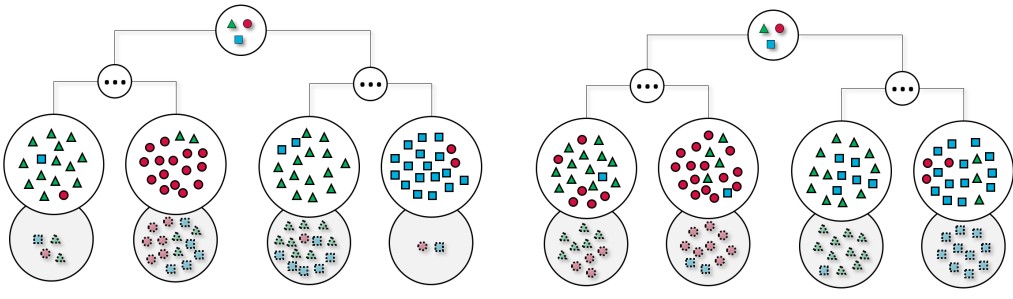

(a) Without domain alignment       (b) With domain alignment

Figure 3: Visualization of trees learned by the proposed methods. The white and gray circles at the leaf nodes represent the source and target data fallen into the node, respectively. As a tree without domain alignment only considers the labeled source data, the target data distributions in leaf nodes are not even, whereas that with domain alignment generates more uniform splits. Refer to Sec. 3.2 and Fig. 2c.

source and target data are evenly split into the children, as it maximizes the entropy of the children (Eq. 2). Thus $IG_{\mathcal{D}}(n)$ in CoBRF collaboratively enforce the even split of target data also.

Fig. 2c illustrates the effect of co-IG compared to conventional IG in split function selection. The yellow line has higher IG as it segments the source data (colored) better, but co-IG also considers the split of target data (gray). Although the blue line has lower IG than the blue, it separates the target data more evenly; thus, the blue line is chosen as the split function. The resulting decision trees by CoBRF are shown in Fig. 3.

The co-IG is closely related to the adversarial learning of the network backpropagation Long et al. (2015); Ganin et al. (2016). In this framework, $IG_{\mathcal{C}}$ and $IG_{\mathcal{D}}$ can be thought of as the classification and adversarial domain alignment, respectively. Thanks to the domain alignment term (co-IG), the CoBRF learns the robust models even with very noisy or small training data without overfitting. We validate the proposed methods from the moderate challenging condition such as $40\%$ noise data to the very severe condition such as $80\%$ noise or merely $10\%$ training data. Further, we evaluate the open set domain adaptation, which has received attention in recent years, in the following section.

## 4    EXPERIMENTAL RESULTS

### 4.1    EXPERIMENTAL SETTING

We use three domain adaptation datasets such as **Office-31** Saenko et al. (2010), **ImageCLEF-DA**[1] and **Office-Home** Venkateswara et al. (2017). We evaluate algorithms using three challenging protocols: noisy, small training data, and weakly supervised open set domain adaptation. To train the CoBRF, we use 100 trees with a maximum depth of 8. When there is no training data fallen on a node, we prune the tree at that node. The number of randomly selected feature dimension for the SVM training is set to 250. The input feature of SVM is normalized for the stable learning of the hyperplane. We repeat the SVM training 15 times to select the optimal split in each node. Due to the space limitation, only representative results are shown in this section. Refer to the appendix for detailed information of the datasets, metric, and full experimental results.

### 4.2    ABLATION STUDY

We evaluate the effect of components in the CoBRF proposed in this paper. The CoBRF uses the balanced pseudo labeling (mid_pseudo), and the hyperplane shift (h_shift) for even data split. As binary labeling is necessary for hyperplane computation, the pseudo method uses randomly-assigned binary pseudo labels without removing the data to make label sizes equal. Therefore the four

---

[1]https://www.imageclef.org/2014/adaptation

Table 1: Ablation study of components for the split function of random forests without the domain alignment. The experiment is performed on Amazon (A), Webcam (W) and DSLR (D) domains of Office-31 with ResNet-50.

| Hyperplane estimation h_shift | pseudo X | mid_pseudo X | pseudo O | mid_pseudo O |
|---|---|---|---|---|
| Accuracy | 70.6 | 72.1 | 74.3 | **74.6** |

Table 2: The effect of $\lambda$ for domain alignment in the CoBRF. The experiment is performed on Office-31 with ResNet-50.

| $\lambda$ | 0 | 0.001 | 0.01 | 0.1 | **0.5** | 1.0 |
|---|---|---|---|---|---|---|
| Accuracy | 70.3 | 71.4 | 72.3 | 74.0 | **74.6** | 74.5 |

Table 3: The effect of the tree numbers and maximum depth in training the CoBRF. The experiment is performed on 40 % noisy condition of the Office-31 (W->A) with ResNet-50.

| Depth | The number of trees (T) | | | |
|---|---|---|---|---|
| | 5 | 10 | 50 | 100 |
| 6 | 62.2 | 65.4 | 67.5 | 67.7 |
| 7 | 60.2 | 63.5 | 67.1 | 67.8 |
| 8 | 58.9 | 63.6 | 67.5 | **68.0** |
| 9 | 55.6 | 60.0 | 66.2 | 67.6 |

Table 4: Performance comparison of the 60 and 80% **Noisy** and 10% **Small** training data protocol on Office-31, ImageCLEF-DA and Office-Home dataset with ResNet-50.

| Method | Office-31 | | | ImageCLEF-DA | | | Office-Home | | |
|---|---|---|---|---|---|---|---|---|---|
| | Noisy | | Small | Noisy | | Small | Noisy | | Small |
| | 60% | 80% | | 60% | 80% | | 60% | 80% | |
| DAN | 37.6 | 19.8 | 66.8 | 36.3 | 19.2 | 74.4 | 32.1 | 18.4 | 43.8 |
| JAN | 48.7 | 24.6 | 69.7 | 42.4 | 19.7 | 76.6 | 35.6 | 21.6 | 45.3 |
| CDAN+E | 49.8 | 22.0 | 67.5 | 54.5 | 25.0 | 79.6 | 34.0 | 15.1 | 44.2 |
| CoBRF | **65.6** | **44.3** | **74.6** | **67.8** | **32.6** | **79.8** | **56.8** | **46.5** | **51.8** |

combinations of (pseudo, mid_pseudo)×h_shift are tested with Office-31. The baseline is (pseudo + no_h_shift). As shown in Table 1 and 8 of appendix, both balancing the pseudo labels and enforcing even splits by translating hyperplanes improve the performance.

Table 2 shows the effect of the parameter $\lambda$ in co-IG formulation (Eq. 4). It confirms that optimizing for the cobalanced distribution helps the alignment of domain distributions. Table 3 also represents the effect of hyperparameters in training CoBRF. The accuracy of the maximum depth 8 with 100 decision trees shows the best result.

## 4.3 NOISY DATA

In this experiment, the training labels of the specified portion of the source domain are randomly changed for the noise condition, which is also referred to as the label corruption in Shu et al. (2019). Corruption levels are set to $40, 60$ and $80\%$ of the source domain (refer to the appendix for full experimental results).

We conduct noisy conditions for the Office-31, ImageCLEF-DA and Office-Home datasets in Table 4 and 5. We test DAN Long et al. (2015), JAN Long et al. (2017), and CDAN+E Long et al. (2018)

Table 5: Performance comparison of the 40% **Noisy** protocol on Office-31 with ResNet-50.

| Method | Domain adaptation | | | | | | |
|---|---|---|---|---|---|---|---|
| | A→D | A→W | D→A | D→W | W→A | W→D | Average |
| RTN Long et al. (2016) | 76.1 | 64.6 | 49.0 | 71.7 | 56.2 | 82.7 | 66.7 |
| ADDA Tzeng et al. (2017) | 61.2 | 61.5 | 45.5 | 65.1 | 49.2 | 74.7 | 59.5 |
| MentorNet Jiang et al. (2018) | 75.0 | 74.4 | 43.2 | 70.6 | 54.2 | 85.9 | 67.2 |
| TCL Shu et al. (2019) | **83.3** | 82.0 | 60.5 | 77.2 | 65.7 | 90.8 | 76.6 |
| CoBRF | 81.9 | **82.1** | **65.4** | **81.1** | **68.0** | **92.8** | **78.5** |

Table 6: Performance comparison of the **OpenSet1** protocol on the Office-31 dataset with AlexNet Krizhevsky et al. (2012).

| Method | A→D | A→W | D→A | D→W | W→A | W→D | Average |
|---|---|---|---|---|---|---|---|
| OSVM | 59.6 | 57.1 | 14.3 | 44.1 | 13.0 | 62.5 | 40.6 |
| ATI-$\lambda$ + OSVM | 72.0 | 65.3 | 66.4 | 82.2 | 71.6 | 92.7 | 75.0 |
| Saito et al. (2018) | 76.6 | 74.9 | 62.5 | 94.4 | **81.4** | **96.8** | 81.1 |
| CoBRF | **86.0** | **80.5** | **73.0** | **94.5** | 69.4 | 94.6 | **83.0** |

Table 7: Performance comparison of the **OpenSet2** protocol on the Office-31 dataset with ResNet-50. Results of CoBRF* are from a more challenging setup. Refer to Sec. 4.5.

| Method | A ↔ D | A ↔ W | D ↔ W | Average |
|---|---|---|---|---|
| JAN Long et al. (2017) | 65.5 | 63.8 | 74.7 | 68.0 |
| ATI-semi Panareda Busto & Gall (2017) | 72.0 | 73.4 | 77.8 | 74.7 |
| CDA Tan et al. (2019) | 75.2 | 77.1 | 88.1 | 80.1 |
| CoBRF | **82.3** | **83.1** | **92.9** | **86.1** |
| CoBRF* | 82.0 | 81.2 | 89.7 | 84.3 |

algorithms[2] on the same noisy condition for comparison. Table 5 shows the result of 40% noisy training data for Office-31 with ResNet-50 He et al. (2016). The proposed CoBRF outperforms all other algorithms in average accuracy. The result confirms that the CoBRF improves the performance in most settings, and interestingly, Table 4 shows the more severe the noise is, the larger the performance improvement gets.

### 4.4 SMALL TRAINING DATA

In this experiment, we use only 10% of training samples to evaluate the performance against overfitting. We perform the experiments on Office-31, Office-Home, and ImageCLEF-DA datasets with ResNet-50. The result of Table 4 shows the CoBRF achieves favorable performance compared to the other algorithms. Full experimental results are presented in the appendix.

### 4.5 OPEN SET EXPERIMENTS

We perform two open set evaluation protocols proposed in Saito et al. (2018); Tan et al. (2019).

**OpenSet1:** The first open set protocol Saito et al. (2018) uses 11 classes (10 known and 1 unknown) of the Office-31 dataset. The labels from 1 to 10 of both source and target domains are marked as the known class, and all data with label 21∼31 in the target domain are used as one unknown class. According to Saito et al. (2018) the unknown class of the source data is not used in training, and the unknown class of the target data is classified by thresholding the class probability. The thresholding value is set to 0.3. Table 6 shows the result of CoBRF as well as the state-of-the-art methods. The

---

[2]DAN, JAN: https://github.com/thuml/Xlearn , CDAN+E: https://github.com/thuml/CDAN

CoBRF achieves the best performance among all algorithms on Office-31. It also demonstrates the effectiveness of the proposed method under the challenging adaptation condition.

**OpenSet2:** Recently, another open set protocol is proposed in Tan et al. (2019), which uses partially overlapping known classes between the source and target domain. Each domain has 5 known-and-common classes, 5 known-but-different classes, and 1 unknown class for all other training data, thus in total there are 15 known and 1 unknown classes. First, according to Tan et al. (2019), 3 samples per class per domain and 9 samples in the unknown class per domain are used in training. Hence the total number of training samples is (3 samples $\times$ 10 classes/domain + 9 samples_in_unknown) $\times$ 2 domains = 78. All other algorithms and CoBRF results in Table 7 are using this protocol. Additionally, we evaluate more challenging setup, where the training data are sampled regardless of the domain, i.e., the data in common classes (including unknown) are merged before being sampled. In this case, 3 samples $\times$ 15 classes + 9 samples_in_unknown = 54 in total are used. The results in CoBRF* rows are acquired in this setup. We confirm that CoBRF works well compared to state-of-the-art methods under the OpenSet2 and more challenging condition.

## 5 CONCLUSION

We propose a novel cobalanced random forest (CoBRF) algorithm for challenging conditions and open set protocols. The CoBRF enhances the discriminative ability of the random forest by building balanced decision trees by the even split. The proposed CoBRF algorithm also employs the adversarial learning for domain alignment and benefits the effectiveness against the overfitting to the labeled source data. We extensively evaluate the proposed algorithms using challenging experimental protocols and demonstrate its superior performance over the baseline and state-of-the-art methods.

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

# Appendix

## A  DATASET AND METRIC

We use three domain adaptation datasets for the experiments.

**Office-31** is the domain adaptation dataset, which consists of three domains, Amazon (A), Webcam (W) and DSLR (D). The dataset has 4,652 images with 31 categories, and we evaluate all combinations of the domains transfer following the previous work.

**ImageCLEF-DA** includes 12 classes in each domain with 50 images per category per domain. The three domains of Caltech-256 (C), ImageNet ILSVRC 2012(I) and Pascal VOC 2012 (P) are used for evaluation.

**Office-Home** with 15,500 images and 65 categories is a larger dataset than office-31. We use four domains of Artistic (A), Clip Art (C), Product (P) and Real-World (R) for evaluation.

We follows the standard metric for the experiments. The averaged accuracy over the classes Saito et al. (2018); Tan et al. (2019) is used to OpenSet1 and OpenSet2, and for other experiments the classification accuracy is used as the metric.

## B  ABLATION STUDY

The full tables of ablation study in the manuscript are presented.

Table 8: Ablation study of components in CoBRF.

| Hyperplane estimation | h_shift | Domain adaptation | | | | | | |
|---|---|---|---|---|---|---|---|---|
| | | A→D | A→W | D→A | D→W | W→A | W→D | Average |
| pseudo | X | 73.3 | 73.6 | 57.3 | 81.4 | 53.7 | 84.4 | 70.6 |
| mid_pseudo | X | 74.4 | 75.0 | 58.3 | 81.2 | 57.2 | 86.8 | 72.1 |
| pseudo | O | **76.4** | **78.1** | 58.1 | 82.8 | 61.0 | 89.4 | 74.3 |
| mid_pseudo | **O** | 76.1 | 77.3 | **59.0** | **83.4** | **62.0** | **89.6** | **74.6** |

Table 9: The effect of $\lambda$ in CoBRF. The experiment is performed on Office-31 with ResNet-50.

| $\lambda$ | Domain adaptation | | | | | | |
|---|---|---|---|---|---|---|---|
| | A→D | A→W | D→A | D→W | W→A | W→D | Average |
| 0 | 76.8 | 75.3 | 51.7 | 75.2 | 55.1 | 87.7 | 70.3 |
| 0.001 | 76.2 | 76.3 | 51.7 | 78.3 | 57.3 | 88.4 | 71.4 |
| 0.01 | 75.9 | 76.5 | 52.9 | 80.8 | 58.7 | 88.9 | 72.3 |
| 0.1 | **77.4** | 76.9 | 57.6 | 81.7 | 61.0 | 89.2 | 74.0 |
| **0.5** | 76.1 | **77.3** | 59.0 | 83.4 | **62.0** | **89.6** | **74.6** |
| 1.0 | 76.0 | 77.3 | **59.9** | **83.6** | 60.9 | 89.2 | 74.5 |

## C  Noisy Data

The full tables of noisy protocol in the manuscript are presented.

Table 10: Performance comparison of the 60 and 80% **Noisy** protocol on Office-31 with ResNet-50.

| Method | A→D | | A→W | | D→A | | D→W | | W→A | | W→D | | Average | |
|---|---|---|---|---|---|---|---|---|---|---|---|---|---|---|
| | 60 | 80 | 60 | 80 | 60 | 80 | 60 | 80 | 60 | 80 | 60 | 80 | 60 | 80 |
| DAN | 44.8 | 23.2 | 39.5 | 22.1 | 17.5 | 12.0 | 49.4 | 22.8 | 20.4 | 13.5 | 53.9 | 25.2 | 37.6 | 19.8 |
| JAN | 62.2 | 29.7 | 61.6 | 32.6 | 28.4 | 17.4 | 56.3 | 26.1 | 28.0 | 14.1 | 55.4 | 27.8 | 48.7 | 24.6 |
| CDAN+E | 56.7 | 24.4 | 59.0 | 26.0 | 34.4 | 13.3 | 55.1 | 24.0 | 35.8 | 17.4 | 57.6 | 27.3 | 49.8 | 22.0 |
| CoBRF | **78.0** | **68.6** | **79.5** | **68.5** | **49.7** | **28.4** | **58.4** | **29.3** | **58.2** | **35.7** | **69.8** | **35.5** | **65.6** | **44.3** |

Table 11: Performance comparison of the 60 and 80% **Noisy** protocol on ImageCLEF-DA with ResNet-50.

| Method | C→I | | C→P | | I→C | | I→P | | P→C | | P→I | | Average | |
|---|---|---|---|---|---|---|---|---|---|---|---|---|---|---|
| | 60 | 80 | 60 | 80 | 60 | 80 | 60 | 80 | 60 | 80 | 60 | 80 | 60 | 80 |
| DAN | 33.3 | 19.2 | 25.8 | 15.6 | 42.9 | 29.1 | 35.1 | 17.3 | 40.2 | 16.3 | 40.8 | 17.9 | 36.3 | 19.2 |
| JAN | 39.6 | 18.9 | 33.6 | 15.3 | 49.2 | 28.9 | 38.6 | 17.3 | 50.8 | 19.7 | 42.5 | 18.1 | 42.4 | 19.7 |
| CDAN+E | 57.2 | 24.2 | 41.8 | 19.0 | 68.1 | 34.0 | 52.3 | 23.1 | 52.9 | 26.3 | 55.0 | 23.7 | 54.5 | 25.0 |
| CoBRF | **72.4** | **34.4** | **60.1** | **30.2** | **73.8** | **39.8** | **61.3** | **31.7** | **71.4** | **31.0** | **67.8** | **28.7** | **67.8** | **32.6** |

Table 12: Performance comparison of the 60 and 80% **Noisy** protocol on Office-Home with ResNet-50.

| Method | Ar→Cl | | Ar→Pr | | Ar→Rw | | Cl→Ar | | Cl→Pr | | Cl→Rw | | Pr→Ar | | Pr→Cl | | Pr→Rw | | Rw→Ar | | Rw→Cl | | Rw→Pr | | Average | |
|---|---|---|---|---|---|---|---|---|---|---|---|---|---|---|---|---|---|---|---|---|---|---|---|---|---|---|
| | 60 | 80 | 60 | 80 | 60 | 80 | 60 | 80 | 60 | 80 | 60 | 80 | 60 | 80 | 60 | 80 | 60 | 80 | 60 | 80 | 60 | 80 | 60 | 80 | 60 | 80 |
| DAN | 11.0 | 5.2 | 36.2 | 18.6 | 43.3 | 22.8 | 28.4 | 18.2 | 42.9 | 22.9 | 41.5 | 23.5 | 29.8 | 15.9 | 21.3 | 13.6 | 23.6 | 13.1 | 38.4 | 23.6 | 15.2 | 9.6 | 54.3 | 33.1 | 32.1 | 18.4 |
| JAN | 21.9 | 10.6 | 40.6 | 21.2 | 44.5 | 22.3 | 31.6 | 18.1 | 43.1 | 24.7 | 43.2 | 25.4 | 31.0 | 17.4 | 23.3 | 14.8 | 49.3 | 29.7 | 16.8 | 24.4 | 26.1 | 16.0 | 56.1 | 34.2 | 35.6 | 21.6 |
| CDAN+E | 22.2 | 9.8 | 41.7 | 20.7 | 49.4 | 13.9 | 24.9 | 11.4 | 38.5 | 17.1 | 37.9 | 16.6 | 23.9 | 10.8 | 18.6 | 8.8 | 40.5 | 17.4 | 33.3 | 16.2 | 23.8 | 10.3 | 52.6 | 27.6 | 34.0 | 15.1 |
| CoBRF | **38.1** | **26.2** | **56.3** | **38.9** | **64.6** | **44.9** | **51.5** | **41.2** | **59.1** | **47.2** | **61.0** | **50.2** | **53.1** | **46.9** | **43.1** | **36.8** | **71.3** | **63.2** | **62.7** | **57.0** | **46.7** | **40.5** | **73.7** | **65.2** | **56.8** | **46.5** |

## D  Small Training Data

The full tables of small training data protocol in the manuscript are presented.

Table 13: Performance comparison of the 10% **Small** training sample protocol on Office-31 with ResNet-50.

| Method | A→D | A→W | D→A | D→W | W→A | W→D | Average |
|---|---|---|---|---|---|---|---|
| DAN | 69.7 | 69.8 | 48.7 | 74.2 | 53.7 | 83.9 | 66.8 |
| JAN | 74.7 | 76.7 | 49.8 | 76.6 | 55.4 | 85.2 | 69.7 |
| CDAN+E | **77.8** | 76.6 | 42.3 | 75.0 | 49.5 | 83.6 | 67.5 |
| CoBRF | 76.1 | **77.3** | **59.0** | **83.4** | **62.0** | **89.6** | **74.6** |

Table 14: Performance comparison of the 10% **Small** training sample protocol on ImageCLEF-DA with ResNet-50.

| Method | C→I | C→P | I→C | I→P | P→C | P→I | Average |
|---|---|---|---|---|---|---|---|
| DAN | 79.4 | 65.9 | 86.0 | 67.5 | 74.3 | 73.2 | 74.4 |
| JAN | 81.3 | 71.0 | 89.5 | 69.0 | 73.8 | 75.3 | 76.6 |
| CDAN+E | **87.8** | 70.8 | **91.3** | 71.3 | 79.8 | 76.9 | 79.6 |
| CoBRF | 83.2 | **72.2** | 90.1 | **74.1** | **80.7** | **78.6** | **79.8** |

Table 15: Performance comparison of the 10% **Small** training sample protocol on Office-Home with ResNet-50.

| Method | A→C | A→P | A→R | C→A | C→P | C→R | P→A | P→C | P→R | R→A | R→C | R→P | Average |
|---|---|---|---|---|---|---|---|---|---|---|---|---|---|
| DAN | 22.3 | 38.5 | 47.9 | 35.3 | 44.9 | 46.4 | 44.3 | 32.5 | 62.8 | 54.1 | 33.9 | 63.2 | 43.8 |
| JAN | 22.3 | 38.2 | 49.3 | 38.4 | 48.3 | 49.0 | 46.0 | 32.3 | 65.1 | 55.4 | 32.1 | 66.8 | 45.3 |
| CDAN+E | 21.5 | 35.1 | 46.2 | 36.0 | 45.3 | 46.2 | 45.0 | 33.4 | 63.0 | 56.6 | 33.2 | 69.7 | 44.2 |
| CoBRF | **32.2** | **49.3** | **55.3** | **45.0** | **53.5** | **55.3** | **52.4** | **39.8** | **67.9** | **59.0** | **42.6** | **69.8** | **51.8** |

