# OpenReview forum: "Collaborative Training of Balanced Random Forests for Open Set Domain Adaptation"
_ICLR.cc/2020/Conference — Reject_

### Official Review · AnonReviewer3 · 2019-10-22
**Official Blind Review #3**

**Rating:** 3

**Review:**

Summary:
This paper introduces a method for domain adaptation, where each domain has noisy examples. Their method is based on a decision tree in which the data at each node are split into equal sizes while maximizing the
information gain.  They also proposed a way to reduce domain alignment. Their method is tested on several noisy domain adaptation settings and performs better than other baseline methods.

Pros:
Their idea to utilize a decision tree for domain adaptation sounds novel.
Experiments indicate the effectiveness of their method.

Cons:
This paper is not well-written and has many unclear parts.
1, The presentation of the problem set is unclear throughout this paper. In the abstract, they mentioned that they tackle the situation where both source and target domains contain noisy examples. However, they did not define the exact problem setting in any section. I could not understand what kind of problem setting motivated their method, which makes it hard to understand their method.
2, How they actually optimized the model is also unclear. From Eq 1~4, it is hard to grasp how they trained the model.
3, In open-set domain adaptation, simply minimizing domain-distance can harm the performance. How does the method avoid this issue? It was also unclear.
4, Experimental setting seems to be wrong and unclear. In Openset1, they say that "The labels from 1 to 10 of both source and target domains are marked as the known class, and all data with label 11∼20 in the source domain and label 21∼31 in the
target domain are used as one unknown class". However, Saito et al. (2018) used 21-31 classes in the target domain as one unknown class. In addition, "According to Saito et al. (2018) the target data of the unknown class is not used in training, ", they used the 21-31 classes for training in an unsupervised way. How is this method used to detect unknown class? Is there any threshold value set for it?
5, The experimental setting is unclear. In 4.4, ", we use only 10% of training samples", does it mean 10 % training source examples or target examples? This setting is also unclear.

From the cons written above, this paper has too many unclear parts in the experiments and method section. I cannot say the result is reproducible given the content of the paper and the result is a reliable one. They need to present more carefully designed experiments.


**Experience Assessment:**

I have published in this field for several years.

**Review Assessment: Checking Correctness Of Derivations And Theory:**

I assessed the sensibility of the derivations and theory.

**Review Assessment: Checking Correctness Of Experiments:**

I carefully checked the experiments.

**Review Assessment: Thoroughness In Paper Reading:**

I read the paper at least twice and used my best judgement in assessing the paper.

---

> ### Author Response · Authors · 2019-11-12
> **The answer to Reviewer #3**
>
> Thank you for helpful comments on our paper.
>
> 1. We use noisy labels only for the ‘source’ domain in the experiments since we design this experiment to validate the robustness of the proposed algorithm.
> We randomly change the original label to create a noisy setting. The specified portion of changed noise data is 40% in Table 5 and 60% and 80% in Table 4. This noisy setting is also described as label corruption in [1].
> Accordingly, we revised the paper to introduce this detail in page 7, as below.
>
> —————————————— Revised paper ————————————————————-
> In this experiment, the training labels of the specified portion of the source domain are randomly changed for the noise condition, which is also referred to as the label corruption in Shu et al. (2019).
> Corruption levels are set to $40, 60$, and $80\%$ of the source domain (refer to the supplementary material for full experimental results.
> ——————————————————————————————————————————--
>
> [1] Yang Shu, Zhangjie Cao, Mingsheng Long, and Jianmin Wang. Transferable curriculum for weakly supervised domain adaptation. In AAAI Conference on Artificial Intelligence, 2019
>
>
> 2. We revised the paper to describe the training process of the proposed method more clearly. As mentioned by R2, the hyperparameters in training the CoBRF and ablation studies are added in page 6 and 7 as below.
>
> ————————--Revised paper ——————————————————————————-
> To train the CoBRF, we use 100 trees with a maximum depth of 8.
> When there is no training data fallen on a node, we prune the tree at that node.
> The number of randomly selected feature dimension for the SVM training is set to 250.
> The input feature of SVM is normalized for the stable learning of the hyperplane.
> We repeat the SVM training 15 times to select the optimal split in each node.
>
>    Depth | The number of trees (T)
>                |  5 | 10 | 50 | 100 |
>       6   | 62.2 | 65.4 | 67.5 | 67.7 |
>       7   | 60.2 | 63.5 | 67.1 | 67.8 |
>       8   | 58.9 | 63.3 | 67.5 | 68.0 |
>       9   | 55.6 | 60.0 | 66.2 | 67.6 |
> ——————————————-—-——————————————————————————-
>
>
> 3. Yes, for open set domain adaptation, we need to elaborate train models so that it does not simply minimize or adjust domain shifts. The source and target domains in the open set condition have a different set of classes, including unknown classes. Therefore, due to this condition of the open set domain adaptation, the overfitting problem should be suppressed. To do this, we take advantage of the good property of random forests [2], which rarely overfit to the training data since they form weak ensembles of multiple decision trees as studied in previous works [3][4]. Therefore, we argue that the proposed CoBRF is robust to the overfitting problem due to the unbalanced classes and the existence of the unknown class of the open set domain adaptation task.
>
> [2] Leo Breiman. Random forests. Machine Learning, 45(1):5–32, 2001.
> [3] Abraham J Wyner, Matthew Olson, Justin Bleich, and David Mease. Explaining the success of adaboost and random forests as interpolating classifiers. The Journal of Machine Learning Research, 18(1):1558–1590, 2017.
> [4] Heitor M Gomes, Albert Bifet, Jesse Read, Jean Paul Barddal, Fabrício Enembreck, Bernhard Pfharinger, Geoff Holmes, and Talel Abdessalem. Adaptive random forests for evolving data stream classification. Machine Learning, 106(9-10):1469–1495, 2017.
>
> 4. First of all, we apologize for missing the detailed description of the Openset1 experiment. We follow the setting of Saito et al. (2018) [5], which do not use 21-31 classes of the source data for training, and thus only 1-10 classes of source domain are included in the training step.
> We detect the unknown class by thresholding the estimated probability of the CoBRF for target data. The threshold value is set to 0.3 in the experiments.
>
> We revised the paper as follows, and you can find the revised version on page 8 of the paper.
>
> —————————————— Revised paper ————————————————————-
> all data with label 21$\sim$31 in the target domain are used as one unknown class. According to Saito et al. (2018), the unknown class of the source data is not used in training, and the unknown class of the target data is classified by thresholding the class probability. The thresholding value is set to 0.3 in the epxeriments.
>
> ————————————————————---————————————————————-
>
> [5] Kuniaki Saito, Shohei Yamamoto, Yoshitaka Ushiku, and Tatsuya Harada. Open set domain adaptation by backpropagation. In European Conference on Computer Vision, pp. 153–168, 2018.
>
> 5. We use 10% of source data and all (100%) of target data. We validate the performance of CoBRF on the small labeled training data (source domain). We also train other baseline methods with the same setting in the experiments for the fair comparison.

---

### Official Review · AnonReviewer1 · 2019-10-23
**Official Blind Review #1**

**Rating:** 3

**Review:**

This paper proposes a new target objects for training random forests that has better generalizability across domains. The authors demonstrated that the proposed method outperforms existing adversarial learning based domain adaptation methods.


Strength

The paper is clearly-written. The two objectives(balanced split and common split distribution between source and target domain) are well motivated and explained in the paper.

The authors show that empirically the proposed method outperform several existing adversarial learning based domain adaptation methods.


Weakness

One of the main draw back of the method is that it relies on the features extracted from existing pre-trained neural networks, and cannot be used to update the representation of the neural networks. While the adversarial learning based method could do end to end training.

It would be great if the authors could clarify the setup of the baseline methods(e.g. Whether the baseline methods also take benefit of imagenet dataset, and is trained end to end).

What will happen if you do not have the imagenet models and have to train all the models from scratch?

Overall I think it is a borderline paper that might be interesting to some audiences in the conference.


**Experience Assessment:**

I have published one or two papers in this area.

**Review Assessment: Checking Correctness Of Derivations And Theory:**

I assessed the sensibility of the derivations and theory.

**Review Assessment: Checking Correctness Of Experiments:**

I carefully checked the experiments.

**Review Assessment: Thoroughness In Paper Reading:**

I read the paper thoroughly.

---

> ### Author Response · Authors · 2019-11-12
> **The answer to Reviewer #1**
>
> Thank you for helpful comments on our paper.
>
> 1. Taking advantage of the ImageNet dataset
> All baseline methods in this paper use pretrained models from the ImageNet dataset.
> For example, OSVM, ATI-λ + OSVM and Saito et al. in Table 6 utilize AlexNet pretrained by the ImageNet dataset. JAN, ATI-semi, and CDA in Table 7 also select ResNet-50 that were pretrained with the ImageNet dataset as their backbone networks.
>
> We also use AlexNet for Table 6 and ResNet-50 for the other tables where two backbone networks are pretrained by ImgaeNet.
>
> Thus both the proposed CoBRF and state-of-the-art baseline works take advantage of the ImageNet dataset to train domain adaptation methods.
>
>
> 2. End-to-end learning of neural networks
> Please note that the main focus of CoBRF is to improve the last classification layer in the neural network assuming that the deap features are fixed.
>
> Although CoBRF does not update the neural network parameters, it outperforms the state-of-the-art works in various experiments. We argue that the pre-trained network provides generalized features for generic classification, which makes CoBRF work well for domain adaptation tasks. Also, most state-of-the-art works learn their models on the pretrained networks from ImageNet, which indicates that they also depend on the capabilities of pretraining on the large dataset.
>
> 3. Training from scratch
> Although CoBRF is mainly for learning more robust and generic classifier, we can train a network with CoBRF from scratch by combining another domain adaptation method. We first train a deep neural network with another method from scratch, and then we can construct CoBRF on the feature set for the trained deep neural networks. In addition, we can train a deep neural network with triplet sampling from the split result of random forests proposed in [1]. Using this method, a deep neural network can be trained from scratch with random forests.
> [1] Under review. Submitted to International Conference on Learning Representations, 2020
>
> We will revise our paper based on the answer in the final version if you give the opportunity to present the paper to the conference.

---

### Official Review · AnonReviewer2 · 2019-10-27
**Official Blind Review #2**

**Rating:** 6

**Review:**

This paper proposes an approach to building random forests that are
balanced in such a way as to facilitate domain adaptation. The authors
propose to split nodes not only based on the Information Gain, but
also so that the sizes of each set passed to left and right children
are equal. Another extension to the standard random forest training
procedure is the use of a collaborative term subtracted from the
information gain over the source domain. This term encourages
alignment of the source and target domains in the leaves of trees in
the forest. Experimental results are given on a range of standard
and open-set domain adaptation datasets.

The paper has a number of issues:

1. There are some problems with clarity, and the English is somewhat rough
   throughout. These problems are not terribly distracting, but the
   manuscript could use more polish.
2. I don't see a detailed discussion anywhere about the
   hyperparameters used for fitting the random forests. How many trees
   are used? What is the max depth? These parameters should be
   discussed and included in the ablations in order to appreciate the
   complexity/performance tradeoffs.

This paper has some interesting ideas in it, and the experimental
results are excellent. I would encourage the authors to move salient
material from the supplementary material to the main article and to
provide a more thorough discussion of the complexity of the models
(the structural parameters of the trees/forests).

**Experience Assessment:**

I have read many papers in this area.

**Review Assessment: Checking Correctness Of Derivations And Theory:**

I assessed the sensibility of the derivations and theory.

**Review Assessment: Checking Correctness Of Experiments:**

I assessed the sensibility of the experiments.

**Review Assessment: Thoroughness In Paper Reading:**

I made a quick assessment of this paper.

---

> ### Author Response · Authors · 2019-11-12
> **The answer to Reviewer #2**
>
> Thank you for helpful comments on our paper.
>
> 1. We will revise the paper to improve readability. We will do our best to refine the rough expressions of the paper. We are working on improving the paper, and we will make it better for the final version. It would be greatly appreciated if more detailed comments could be provided.
>
> 2. We added hyperparameter settings such as the number of trees, maximum depth, feature dimensionality of the SVM training, and the number of repeats in training a random forest to page 6. We also supplemented the ablation study with regard to the number of trees and maximum depths in Table 3. In the ablation study, the maximum depth is 8 with 100 decision trees to consider both accuracy and complexity.
>
> ————————--Revised paper ——————————————————————————-
> To train the CoBRF, we use 100 trees with a maximum depth of 8.
> When there is no training data fallen on a node, we prune the tree at that node.
> The number of randomly selected feature dimension for the SVM training is set to 250.
> The input feature of SVM is normalized for the stable learning of the hyperplane.
> We repeat the SVM training 15 times to select the optimal split in each node.
>
>    Depth | The number of trees (T)
>                |  5 | 10 | 50 | 100 |
>       6   | 62.2 | 65.4 | 67.5 | 67.7 |
>       7   | 60.2 | 63.5 | 67.1 | 67.8 |
>       8   | 58.9 | 63.3 | 67.5 | 68.0 |
>       9   | 55.6 | 60.0 | 66.2 | 67.6 |
> ——————————————-—-——————————————————————————-
>
> Please refer to page 6, 7, and Table 3 for more information on the hyperparameters and ablation study.

---

### Decision · Program_Chairs · 2019-12-19

**Decision:**

Reject

**Comment:**

This paper proposes new target objectives for training random forests for better cross-domain generalizability.

As reviewers mentioned, I think the idea of using random forests for domain adaptation is novel and interesting, while the proposed method has potential especially in the noisy settings. However, I think the paper can be much improved and is not ready to publish due to the following reviewers' comments:

- This paper is not well-written and has too many unclear parts in the experiments and method section. The results are not guaranteed to be reproducible given the content of the paper. Also, the organization of the paper could be improved.

- The open-set domain adaptation setting requires more elaboration. More carefully designed experiments should be presented.

- It remains unclear how the feature extractors can be trained or fine-tuned in the DNN + tree architecture. Applying trees to high-dimensional features sacrifices the interpretability of the tree models, hampering the practical value of the approach.

Hence, I recommend rejection.